# Broadening Discovery through Structural Models: Multimodal Combination of Local and Structural Properties for Predicting Chemical Features.

## Abstract

In recent years, machine learning (ML) has significantly impacted the field of chemistry, facilitating advancements in diverse applications such as the prediction of molecular properties and the generation of molecular structures. Traditional string representations, such as the Simplified Molecular Input Line Entry System (SMILES), although widely adopted, exhibit limitations in conveying essential physical and chemical properties of compounds. Conversely, vector representations, particularly chemical fingerprints, have demonstrated notable efficacy in various ML tasks. Additionally, graph-based models, which leverage the inherent structural properties of chemical compounds, have shown promise in improving predictive accuracy. This study investigates the potential of language models based on fingerprints within a bimodal architecture that combines both graph-based and language model components. We propose a method that integrates the aforementioned approaches, significantly enhancing predictive performance compared to conventional methodologies while simultaneously capturing more accurate chemical information.

## 1 Introduction

The integration of machine learning (ML) has emerged as a transformative force in the natural sciences, particularly in the discipline of chemistry (Chithrananda et al., 2020; Hu et al., 2016; Wang et al., 2022). This integration encompasses various tasks, ranging from the regression of molecular properties, exemplified by quantitative structure-activity relationship (QSAR) models (Wu et al., 2021; Rácz et al., 2021), to complex challenges, such as predicting nuclear magnetic resonance (NMR) spectra from molecular structures (Yao et al., 2023). As an ever-evolving discipline, the latest advancements in machine learning are gradually being adapted for applications in chemistry, albeit with some delay. Molecular representations are fundamental to the application of machine learning in chemistry, and three primary types are typically employed: graph-based (Reiser et al., 2022), string-based (Heller et al., 2015; Weininger, 1988; Krenn et al., 2020), and vector representations (Rogers & Hahn, 2010; Durant et al., 2002).

Graph-based representations conceptualize molecules as molecular graphs, effectively capturing their structural properties. This format naturally aligns with graph neural networks, which have been successfully applied to numerous chemical problems, demonstrating their efficacy in molecular analysis.

String representations, particularly the Simplified Molecular Input Line Entry System (SMILES) (Weininger, 1988), are widely regarded as a standard method for the linear representation of molecular structures. SMILES is typically used for storing compounds in databases and, despite its limitations, effectively represents the structure of a molecule. However, it presents notable shortcomings. Initially designed for efficient storage and representation of molecular data, SMILES lacks comprehensive information regarding the physical and chemical properties of compounds. As machine learning progresses, the inadequacies of SMILES in facilitating in-depth semantic analysis have become increasingly evident.

The other type of representation comprises vector representations, notably fingerprints, which were developed for substructure identification and similarity searching. Numerous studies, such as those presented by (Sabando et al., 2022) and (Wang et al., 2020), have highlighted the effectiveness of fingerprints as molecular representations in machine learning tasks, demonstrating promising results even with conventional algorithms like Support Vector Machines (SVM). However, the potential of fingerprints in the context of modern language models remains largely underexploited.

Recent methods have sought to combine graph models with SMILES-based natural language processing (NLP) transformers (Chithrananda et al., 2020; Ahmad et al., 2022; Wang et al., 2019; Shamshad et al., 2023; Cong et al., 2024), thereby integrating the strengths of both methodologies. Building on this concept, our study aims to develop a transformer-based architecture, which serves as our primary goal. We believe this approach is particularly advantageous due to the transformer's inherent versatility and flexibility, allowing it to address a variety of physicochemical tasks that differ significantly in nature. For example, our architecture could potentially handle tasks such as predicting molecular property values, classifying compounds based on their biological activity, generating novel molecular structures, exploring molecular interactions and complex challenges, such as predicting nuclear magnetic resonance (NMR) spectra from molecular structures or solving the co-crystallisation problem.

Given the diverse challenges presented by these tasks, we seek to create a unified framework that requires only minor modifications for each specific application. By leveraging the strengths of transformer models, we anticipate that our architecture will enable efficient and robust performance across multiple domains within the field of cheminformatics. Ultimately, this comprehensive approach aims to enhance predictive capabilities and foster innovation in drug discovery and molecular design.

The questions we want to tackle with our research are the following:

- *To what extent can a transformer model based on fingerprints, either independently or in conjunction with graph models, enhance performance compared to conventional approaches?*

- *Furthermore, can transformers trained on fingerprint representations improve the quality of multitasking embeddings, thereby providing more robust and nuanced representations that capture the complexities across diverse tasks?*

## 1.1 OUR CONTRIBUTIONS

Despite the advancements made in machine learning applications within chemistry, existing methods, particularly those relying on traditional string representations such as SMILES, exhibit notable shortcomings. These limitations include an inability to comprehensively capture the essential physical and chemical properties of compounds, which hampers their effectiveness in facilitating in-depth semantic analysis. Furthermore, while vector representations like chemical fingerprints have proven effective for certain tasks, their potential within modern language models remains largely unexploited.

Graph neural networks (GNNs) have been effectively utilized to address a variety of challenges within the field of chemistry (David et al., 2020; Kwon et al., 2020). However, many GNNs are specialized for specific tasks and are not inherently designed to generate vector representations of chemical compounds. Several methodologies have been proposed to enhance GNN-based embeddings. For instance, (Hu et al., 2016) introduced two primary concepts: the recovery of masked properties of a molecule, such as the type of a specific atom, and the application of contrastive learning to minimize discrepancies between two subgraphs within a molecule. Additionally, Mol-CLR (Wang et al., 2022) presents a framework based on augmenting molecular graphs by removing atoms, edges, and subgraphs, followed by training a model to reconstruct these components.

In our approach, we it implement more physically accurate atom masking and graph augmentation techniques, enhancing the model's understanding of molecular properties and ensuring a more robust representation of the chemical structure. Furthermore, in the case of the Graphormer, we leverage a more advanced architecture that captures intricate relational patterns within the molecular graphs.

In this paper, we propose a novel methodology[1] that integrates graph-based representations with language models based on fingerprints, effectively addressing these limitations. Our approach encompasses four distinct architectures: the first is a single model based on the RoBERTa framework utilizing Extended Connectivity Fingerprints (ECFP), which serves as a baseline for our investigations. The second and third architectures are bimodal models that pair the RoBERTa model as the language component with graph convolutional networks (GCNs) and Graph Isomorphism Networks (GINs), respectively, utilizing contrastive learning for improved feature extraction. While these smaller models train faster, they may exhibit less potential in complex tasks compared to the fourth architecture, which employs a bimodal structure combining the RoBERTa model with Graphormer, an advanced graph transformer that captures more intricate relational patterns.

In each of the graph models – GCN (Kipf & Welling, 2016), GIN (Xu et al., 2018), and Graphormer (Ying et al., 2021) – we implement a mechanism that synthesizes existing concepts by masking atom and edge features. Each of these models is trained not only to predict these masked features but also to align the embeddings of two augmented versions of the same molecule. This approach reflects a modification of contrastive learning, which remains underutilized in the chemistry domain.

By leveraging the structural attributes of molecules alongside the semantic richness of fingerprints, our innovative bimodal architecture significantly enhances predictive performance and facilitates application across a variety of physicochemical tasks.

## 2 RELATED WORKS

### 2.1 EXTENDED-CONNECTIVITY FINGERPRINTS

Extended-connectivity fingerprints, or shortly ECFP (Rogers & Hahn, 2010), are so-called circular fingerprints that assign a two-dimensional hash array to each molecule. Each element of such an array is a hash corresponding to one atom. It encrypts a fixed set of physical and chemical properties of this atom, such as charge, as well as information about its neighbours.

As applied to our task, there are three significant particularities of ECFP. Firstly, the fingerprint of a single molecule consists of an array of hashes (i.e. in NLP terms, we can think of the array as a sentence and each individual hash as a word). Secondly, each hash is constructed based on a set of physical properties. Thus, each array element is based on physical and chemical data. And lastly, there is a so-called diameter, which shows neighbouring atoms in one iteration. That being said, we cover only those atoms, that are within the diameter's reach. This sensible of surrounding environment representation can be very useful for such tasks as molecular NMR spectroscopy, where chemical environment plays crucial role in spectrum definition.

Over the last few years, a number of methods ((Wang et al., 2020; Sturm et al., 2018)) have pointed out the effectiveness of using ECFP as features for training quite simple models to solve various chemical problems.

Several approaches have employed Extended Connectivity Fingerprints as a representation for training data in natural language processing (NLP) algorithms; however, these methods predominantly rely on relatively conventional machine learning techniques. One notable example is Mol2Vec (Jaeger et al., 2018), which implements the Word2Vec algorithm utilizing ECFP data representation.

### 2.2 SMILES-BASED NLP MODELS

Transformers (Vaswani, 2017) were initially introduced to facilitate the generation of vector representations for natural language processing tasks. Since their inception, they have found widespread application across a variety of domains, including speech recognition, medicine, and neuroscience ((Shamshad et al., 2023; Cong et al., 2024)). There have been several efforts to adapt transformers for chemical applications, exemplified by models such as SmilesBERT (Wang et al., 2019), ChemBERTa (Chithrananda et al., 2020), and ChemBERTa-2 (Ahmad et al., 2022).

---

[1]Our code for all experiments is accessible on https://anonymous.4open.science/r/Transformers-for-Molecules-D10E.

Many of these models have been trained on substantial datasets, including ZINC (Irwin et al., 2012) and PubChem (Kim et al., 2023), demonstrating commendable performance in classification and regression tasks across various established chemical benchmarks. By leveraging a more physics-based input format, namely ECFP, and employing one of the most sophisticated language models, we achieved a significant milestone: a large language model (LLM) trained from scratch on two subsets of the PubChem dataset, comprising 2.5 million and 10 million entries, respectively. This model exhibits performance comparable to those trained on the largest datasets within the field.

## 2.3 GRAPH MODELS

Graph neural networks (GNNs) have been effectively utilized to address a variety of challenges within the field of chemistry (David et al., 2020; Kwon et al., 2020). Many GNNs are highly specialized for specific tasks and are not inherently designed for generating vector representations of chemical compounds.

Several methodologies have been proposed to enhance GNN-based embeddings. For instance, (Hu et al., 2016) introduced two primary concepts: the recovery of masked properties of a molecule, such as the type of a specific atom, and the application of contrastive learning to minimize discrepancies between two subgraphs within a molecule. Additionally, MolCLR (Wang et al., 2022) presents a framework based on the augmentation of molecular graphs through the removal of atoms, edges, and subgraphs, followed by the training of a model to reconstruct these components.

In the graph component of our model, we advocate for an approach that synthesizes these concepts and leverages state-of-the-art models. Specifically, we implement a mechanism to mask atom features and edge features in the case of Graphormer (Ying et al., 2021). The model is trained not only to predict these masked features but also to align the embeddings of two augmented versions of the same molecule. This approach represents a modification of contrastive learning, a technique that remains underutilized in the chemistry domain.

Moreover, (Zhu et al., 2023) introduced a bimodal architecture incorporating a BERT-based large language model (LLM) trained on SMILES alongside a GNN as the graphical representation model. In contrast, we propose a distinct language model that is trained on fingerprints, thus providing a more physically informed perspective and an advanced graph model. Additionally, our approach includes notable differences in the final projection and the processing of embeddings derived from both the language and graph models.

## 3 METHOD

### 3.1 ARCHITECTURE OVERVIEW

The proposed model comprises three primary components, as illustrated in Figure 1: the graph model, the language model, and the projection blocks. The language model is designed to accept Extended Connectivity Fingerprint (ECFP) connectives as input, whereas the graph model processes molecular graphs.

The function of the projection blocks is to transform the embeddings generated by the graph and language models from their respective latent spaces into a unified third latent space.

### 3.2 LANGUAGE MODEL

**Tokenizer.** At first, we made an attempt to use hash values from ECFP format as the direct input of the language model. Such an idea was not prosperous because the range of the hash function's outputs (approximately from $-2^{32}$ to $2^{32}$) is very wide to utilize them as tokens for the model's input. In that case, we decided to include a tokenizer in processing. This step allows us to narrow down the range of possible model vocabulary values. As we work with an array of integers interpreted as text, we cannot afford to use a normal tokenizer, which creates tokens out of text. In this regard, we have chosen the Byte-Pair Encoding Tokenizer as shown in Figure 2, which allows the production of tokens from raw bytes. We have trained this tokenizer on the largest dataset we have - PubChem, containing 10M molecules. Such pipeline modification decreases the $vocab\_size$ of the model to not more than $30,522$.

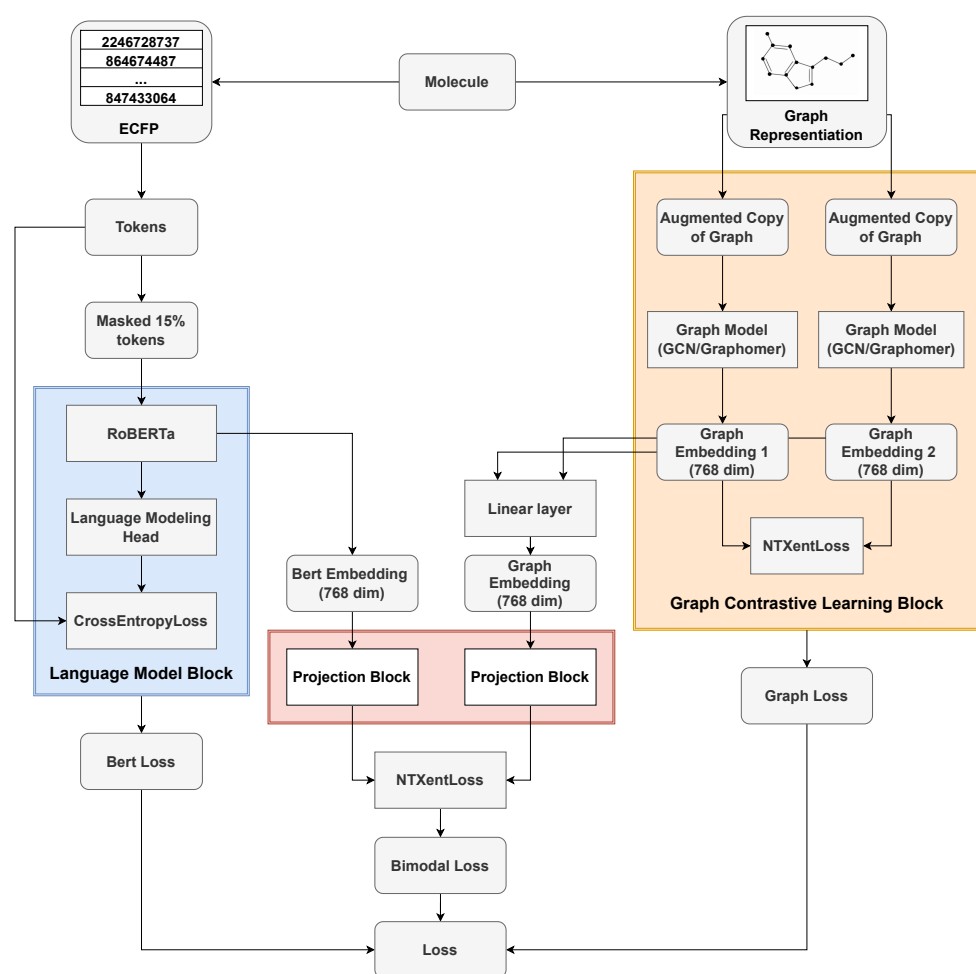

Figure 1: Full architecture of the bimodal model. Language and Graph blocks are grey-outlined.

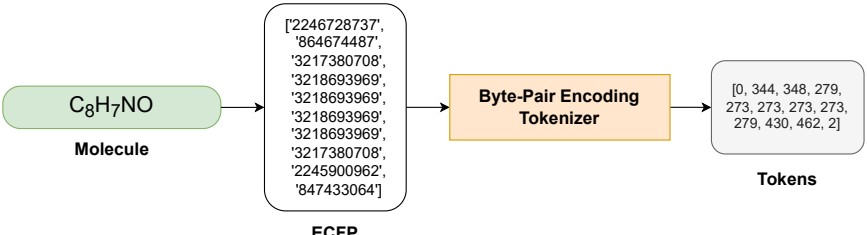

Figure 2: An example of our tokenization process. Tokens "0" and "2" correspond to BOS (begin of sequence) and EOS (end of sequence) respectively.

**RoBERTa training.** We utilize the RoBERTa architecture (Liu, 2019), which has been trained on ECFPs derived from the PubChem and ZINC datasets, as our language model. Within this framework, the encoding of an individual atom in ECFP is interpreted as a "word," while the encoding of an entire molecule is considered analogous to "text." During the training process, the ECFP undergoes standard procedures including the masking of 15% of tokens (representing atom hashes), with the model subsequently predicting the probabilities of these masked tokens. The output embedding is derived from the CLS token located in the penultimate layer of the model.

### 3.3 GRAPH MODEL

**Creation and augmentation of graph.** A graph is constructed from SMILES representations utilizing the RDKit package, wherein each atom is represented as a vertex. Two parameters – atom number and chirality – are designated as attributes of the vertices. In this framework, each bond is represented as an edge, with the bond multiplicity (single, double, triple, or aromatic) serving as the attribute for the edges.

Subsequently, 20% of the atomic attributes are masked, replacing them with a designated mask token. In the case of graphomers, an equivalent approach is applied where 20% of the edge attributes are also masked, transforming these attributes into the mask token.

The augmentation process and the graph model operation scheme are shown in Figure 3.

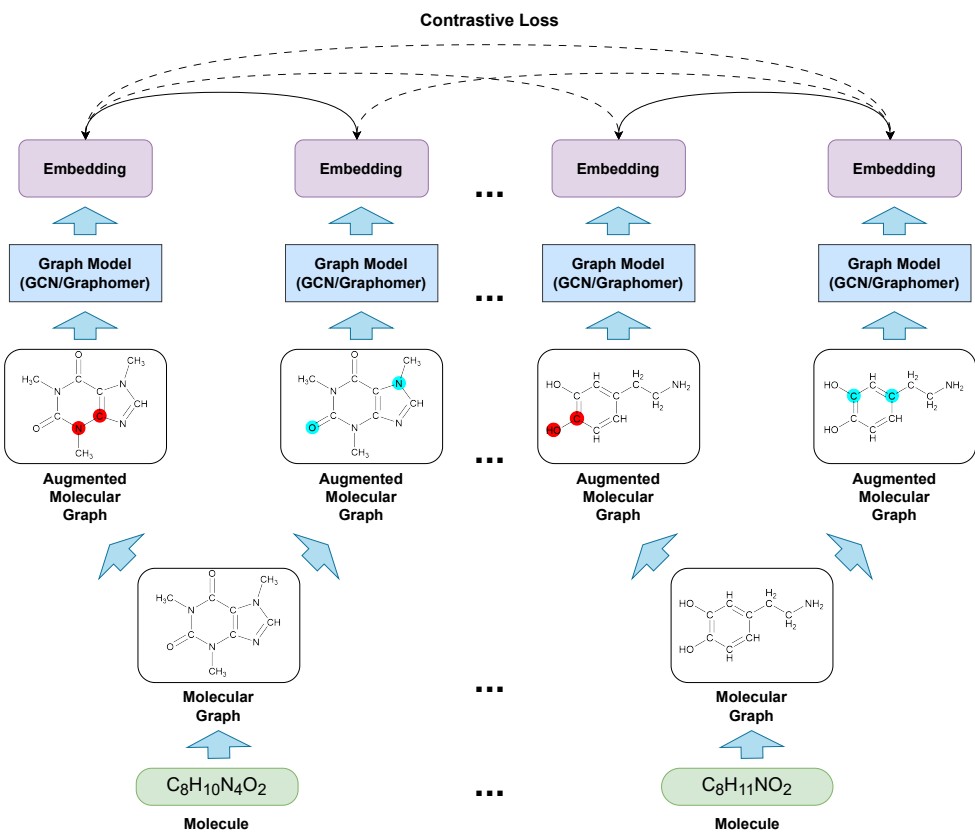

Figure 3: Tops masking process and computing the graph loss for one batch.

**Model training.** In the graph component of our model, we have experimented with three distinct architectures: Graph Isomorphism Network (GIN), Graph Convolutional Network (GCN), and Graphormer. We employ augmentation techniques to transform the molecular graph into two distinct representations. Following this, we train the GCN, GIN, or Graphormer models with the objective of minimizing the differences between the augmentations of one graph and maximize difference between augmentations of different graphs (this process for graphs in one batch is shown in Figure 3).

### 3.4 CONNECTION BETWEEN MODELS

The projection blocks illustrated in Figure 4 of our proposed architecture comprise two linear layers accompanied by two batch normalization blocks. Prior to the application of the final batch normalization block, the ReLU activation function is employed on the embeddings.

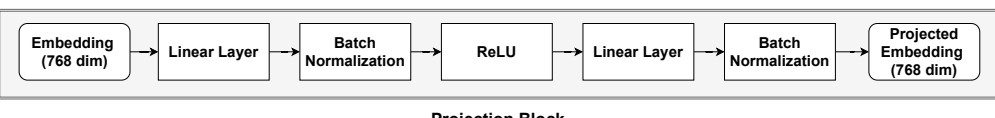

**Projection Block**

Figure 4: The structure of the projection block. It helps to translate output vectors from models to the same linear space.

Let $e_{\text{graph}}$, denote the output of the graph model and $e_{\text{lang}}$ represent the output of the language model. Furthermore, let $\psi_{\text{graph}}$ and $\psi_{\text{lang}}$ be the respective projection blocks for the graph and language models. Define $\mathbb{A}$ as the latent space of the graph model, $\mathbb{B}$ as the latent space of the language model, and $\mathbb{C}$ as the space into which the embeddings are projected. Thus, we have $e_{\text{graph}} \in \mathbb{A}$, $e_{\text{lang}} \in \mathbb{B}$ with $\psi_{\text{graph}} : \mathbb{A} \to \mathbb{C}$ and $\psi_{\text{lang}} : \mathbb{B} \to \mathbb{C}$.

### 3.5 LOSS FUNCTIONS

The loss function used in our model is represented as

$$L = \alpha \cdot L_{\text{lang}} + \beta \cdot L_{\text{graph}} + \gamma \cdot L_{\text{bimodal}}, \tag{1}$$

where $L_{\text{lang}}$ is the loss function of the language model, $L_{\text{graph}}$ is the loss function of the graph part of the model, and $L_{\text{bimodal}}$ is the embedding projection loss function from the graph and language models. Coefficients $\alpha$, $\beta$, and $\gamma$ are some constants which can be considered hyperparameters.

**Language model loss.** $L_{\text{lang}}$ is calculated as regular Cross-Entropy applied to labels and predicted tokens of the language model.

**Graph model loss.** $L_{\text{graph}}$ is defined as NTXent-Loss applied to the batch of augmented graphs' embeddings and to the batch of original graphs' embeddings. It tries to minimize the distance between augmented and original ones of the same index and distances others with different indices.

**NTXent-Loss.** calculates the cosine distance between two vectors and uses the temperature parameter to balance positive and negative pairs. Let $sim(u, v)$ denotes the cosine similarity between vectors $u$ and $v$. Then the loss function for a positive pair of examples (i, j) is as follows:

$$(L_{\text{graph}})_{i,j} = -\log \left( \frac{e^{\text{sim}(u_i, v_j)/\tau}}{\sum_{k=1}^{N} e^{\text{sim}(u_i, v_k)/\tau}} \right), \tag{2}$$

where $N$ is the total number of examples and $\tau$ (temperature) is a parameter that controls the contribution of positive and negative pairs.

**Bimodal Loss.** The bimodal loss, denoted as $L_{\text{bimodal}}$, is defined also as the NT-Xent loss applied to the output embeddings generated by both the language model and the graph model within a given batch. This loss function aims to minimize the distance between the embeddings of the same index from both models while maximizing the distance between embeddings corresponding to different indices.

To achieve this, we employ two distinct projection blocks to map the embeddings from the graph and language models into a unified third latent space. Utilizing a single projection block to project the embeddings from one model into the latent space of the other could inadvertently lead to the training of one model to mimic the behavior of the other. Such an outcome is undesirable, as the distinct functionalities of the models are advantageous for the universal applicability of the bimodal architecture.

### 3.6 SOME ADDITIONAL FEATURES

In this study, we utilize Extended Connectivity Fingerprints (ECFPs) as the data representation for the language model. Unlike SMILES, ECFPs not only encapsulate information pertaining to the

structural design of molecules, but also provide insights into the physical and chemical properties of individual atoms and, importantly, their substructures. As previously discussed, this representation offers a more physically grounded, language-like framework for describing molecules. The inclusion of a defined radius in ECFPs facilitates the adjustment of substructure sizes, which is particularly significant for various chemical tasks, such as predicting NMR spectra or analyzing reaction centers. Consequently, the application of a language model is justified when relevant data reside within small substructures or individual atoms.

Conversely, a graph model comprehensively captures the entire molecular structure, which is advantageous for analyzing extensive connections characterized by numerous substructures. Such scenarios are frequently encountered in biochemistry, particularly when addressing pharmacological compounds or polymer-related challenges. The incorporation of a bimodal architecture that combines both graph and language models enhances the vector representation of molecules in these complex tasks.

In summary, our proposed architecture enables the generation of efficient vector representations for molecules that exhibit significant variability in structure and physicochemical properties.

## 4 EXPERIMENTS

### 4.1 PRETRAINING DATASETS AND DATA PREPARATION

We pretrain our model on parts of two different datasets: PubChem and ZINC (Irwin et al., 2012). Initially, the compounds in them are stored in SMILES format. Then, the data preparation process could be divided into two main parts (as shown in Figure 5).

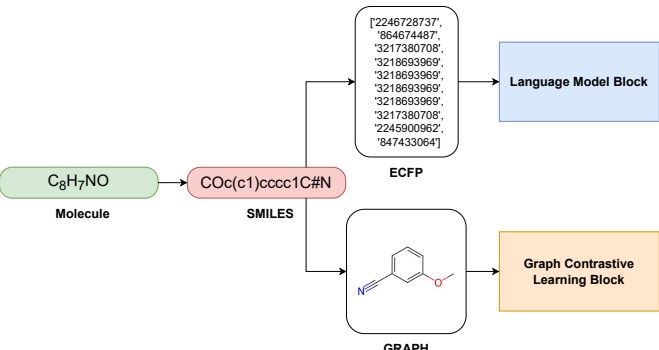

Figure 5: An example of overall molecular data preprocessing. ECFP and Graph representations are generated from the SMILES sequence and are feed-forwarded to the language model and graph model, respectively.

**Language model data.** We construct ECFP from the obtained SMILES of the molecule (the algorithm is given in Appendix A), and then we mask $15\%$ of the elements in the obtained array (having previously performed the tokenization process and considering them as tokens).

**Graph model data.** We build a graph on the SMILES of the molecule and then use the augmentation of it, which transforms it into two different molecule graphs.

The augmentation process consists of masking $20\%$ types of randomly chosen atoms (for GCN and GIN) and both masking $20\%$ of types of randomly chosen atoms and edges (for Graphormer).

We mask only types of atoms and edges, but not the edges and atoms themselves (as in MolCLR approach (Wang et al., 2022)) due to the greater physicochemical validity of this method. For example, if you mask the red highlighted edge in Figure 6 in octyl formate (with CCCCCCCCOC(=O) SMILES-encoding), you get two existing compounds – heptane (CCCCCCC) and methyl formate (COC(=O)). Thus, the model will learn to converge the embeddings of Octyl formate and some total embedding of heptane and methyl formate, which is fundamentally wrong.

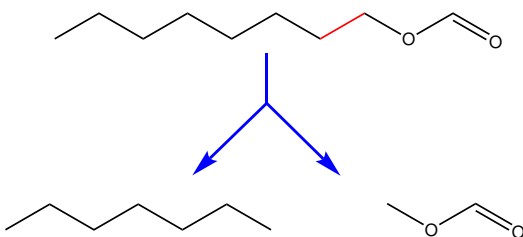

Figure 6: An example of dropping edges problem.

## 4.2 QSAR TASKS

For the zero-shot evaluation of our proposed architecture, we selected a set of widely recognized cheminformatics benchmarks based on the quantitative structure-activity relationship (QSAR) framework. While specially designed descriptors often outperform transformer-based models in these contexts, the simplicity of these benchmarks allows for an assessment of the quality and versatility of our architecture without the confounding influence of the large-scale superstructures typically encountered in more complex problem-solving scenarios.

We evaluated four distinct models: RoBERTa (denoted as ECFP-BERT), which was trained on Extended Connectivity Fingerprints (ECFP); ECFP-BERT in conjunction with Graph Isomorphism Network (GIN); ECFP-BERT combined with Graph Convolutional Network (GCN); and ECFP-BERT integrated with Graphormer. The first three models utilized a dataset comprising 10 million entries sourced from the PubChem database, while the fourth model was trained on a smaller dataset of 1 million entries from the same source.

Several of the utilized datasets – namely, BBBP (Sakiyama et al., 2021), Tox21 (Richard et al., 2020), ClinTox (Wu et al., 2018), BACE (Wu et al., 2018), MUV (Rohrer & Baumann, 2009), and HIV (Pan et al., 2007) – are specifically oriented towards classification tasks. The results for these datasets, along with comparisons to other proposed architectures, are summarized in Table 1. The receiver operating characteristic area under the curve (ROC-AUC) was employed as the evaluation metric.

Conversely, the remaining datasets – QM7 (Blum & Reymond, 2009), (Rupp et al., 2012), QM8 (Ramakrishnan et al., 2015), QM9 (Ruddigkeit et al., 2012a), (Ruddigkeit et al., 2012b), FreeSolv (Mobley & Guthrie, 2014), ESOL (Delaney, 2004), and Lipo – are focused on regression tasks. The mean absolute error (MAE) was utilized as the metric for the QM7, QM8, and QM9 datasets, while the mean squared error (MSE) served as the metric for FreeSolv, ESOL, and Lipo. The findings and comparative analysis with other architectures are presented in Table 2.

Most classification datasets are intrinsically linked to biochemical tasks, often featuring relatively large molecules. It has been observed that language models trained on SMILES representations, such as ChemBERTa, yield only modest performance metrics. This limitation arises from their inability to effectively account for atoms that are situated at significant distances from one another. In contrast, our models exhibit substantial improvements in metric performance when incorporating a graph component, which enhances the capture of molecular structural information.

In the context of regression problems, where smaller molecules are more prevalent, language models demonstrate comparatively strong performance.

Thus, it becomes evident that both components of the architecture – namely the graph and language models – are generally advantageous for achieving optimal model performance.

Graphormer, being a more complex model, tends to exhibit superior performance on large datasets. However, it often struggles with smaller datasets due to insufficient data for effective pre-training. Consequently, we recommend utilizing the BERT+GIN and BERT+GCN models for tasks characterized by limited data availability. Conversely, the BERT+Graphormer architecture is more suitable for intricate tasks that require the establishment of complex internal connections among nodes.

Table 1: Results for classification tasks. ROC-AUC metric (higher is better) for BBBP, Tox21, ClinTox, BACE, MUV and HIV datasets.

| Models | Datasets | | | | | | |
| --- | --- | --- | --- | --- | --- | --- | --- |
| | BBBP | Tox21 (NR-AR) | ClinTox (FDA_APPROVED) | ClinTox (CT_TOX) | BACE | MUV | HIV |
| MolCLR(GCN) (Wang et al., 2022) | 0.72 | 0.70 | 0.66 | 0.69 | 0.71 | 0.67 | 0.78 |
| MolCLR(GIN) (Wang et al., 2022) | 0.74 | 0.74 | 0.87 | 0.77 | **0.81** | 0.57 | 0.76 |
| ChemBERTa (Chithrananda et al., 2020) | 0.64 | 0.75 | - | 0.73 | 0.72 | 0.66 | 0.62 |
| ECFP-BERT (ours) | 0.82 | 0.71 | 0.71 | 0.69 | 0.73 | 0.61 | 0.65 |
| BERT+GIN (ours) | **0.88** | **0.79** | **0.88** | 0.71 | 0.79 | 0.70 | 0.74 |
| BERT+GCN (ours) | 0.85 | 0.79 | 0.71 | 0.69 | 0.73 | 0.64 | 0.73 |
| BERT+Graphormer (ours) | 0.77 | 0.74 | 0.87 | **0.78** | 0.75 | **0.71** | **0.81** |

Table 2: Results regression tasks, MAE (less is better) metric for QM7, QM8 and QM9 datasets. MSE for FreeSolv, ESOL and Lipo.

| Models | Datasets | | | | | |
| --- | --- | --- | --- | --- | --- | --- |
| | QM7 | QM8 (E1-CC2) | QM9 (gap) | FreeSolv | ESOL | Lipo |
| MolCLR(GCN) (Wang et al., 2022) | 85.4 | 0.0178 | 0.0317 | 3.25 | 1.41 | 0.95 |
| MolCLR(GIN) (Wang et al., 2022) | 91.6 | 0.0167 | 0.0225 | 2.88 | 1.25 | 0.65 |
| ChemBERTa (Chithrananda et al., 2020) | 177.2 | - | 0.0317 | 3.47 | 1.48 | 0.71 |
| ECFP-BERT (ours) | 159.4 | 0.0306 | 0.0148 | 2.09 | 1.03 | 0.81 |
| BERT+GIN (ours) | 83.4 | **0.0065** | **0.0060** | **0.35** | **0.27** | **0.31** |
| BERT+GCN (ours) | 84.5 | 0.0092 | 0.0078 | 0.55 | 0.36 | 0.54 |
| BERT+Graphormer (ours) | **81.9** | 0.0291 | 0.0142 | 2.32 | 1.001 | 0.90 |

## 5 CONCLUSION

Our proposed set of architectures, consisting of ECFP RoBERTa (ECFP-BERT) and bimodal configurations that integrate ECFP-BERT as the language branch alongside Graph Convolutional Network (GCN), Graph Isomorphism Network (GIN), or Graphormer as the graph branch, has demonstrated some of the most promising performance metrics compared to existing models in the domain for various quantitative structure-activity relationship (QSAR) problems across a range of well-established benchmarks.

While specialized descriptors typically outperform transformer-based models for these challenges, these benchmarks serve as a simplified context, thereby allowing us to assess the quality and versatility of our architecture without the confounding influence of large-scale superstructures commonly encountered in more complex scenarios.

To further validate our model, we intend to explore additional challenges, including co-crystal prediction, the prediction of nuclear magnetic resonance (NMR) spectra from molecular structure, and other physicochemical tasks. For such demanding tasks, transformer-based architectures often yield significantly superior results compared to straightforward augmentations of task-specific descriptors.

However, it is important to note that addressing these tasks will necessitate considerable modifications to the existing architecture. Consequently, the outcomes will be contingent not only upon the quality of the embeddings currently utilized but also on the enhancements made to the model architecture itself.

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

## A  MORE DETAILS

### A.1  ECFP CONSTRUCTION ALGORITHM

ECFP (Rogers & Hahn, 2010) is a so-called circular fingerprint that assigns a two-dimensional hash array to each molecule using the following algorithm:

1. The initial step is assigning an integer identifier to each atom.

2. The iterative update stage, in which the identifier of each atom is updated with the identifiers of its neighbours.

3. Duplicate removal - a stage in which several occurrences of the same feature are reduced to a single representation in the feature list.

One iteration for a single atom is as follows:

1. An array of integers containing the iteration number and the ID of the given atom is initialized.

2. The attached atoms are sorted in deterministic order using the bond order (single, double, triple, and aromatic) and the current ID of each attached atom. or each attachment, the attachment ID and bond order are added to the array.

3. The array is hashed into a single 32-bit integer. This is the new atom identifier.

### A.2  GRAPH MODELS

**GCN.**  The Graph Convolutional Network (GCN), as introduced by Kipf and Welling (Kipf & Welling, 2016), constitutes a significant advancement in the field of graph neural networks, employing convolutional operations tailored specifically for graph data structures. Distinct from conventional neural networks that utilize linear transformations through a weight matrix $\mathbf{W}$, represented mathematically as $h = \mathbf{W}x$, GCNs incorporate the inherent topological characteristics of the graph to update node representations. This approach is particularly advantageous given the phenomenon of network homophily, wherein connected nodes are more likely to exhibit similar attributes.

GCNs operate through a principle known as neighborhood aggregation, which amalgamates the features of a target node with those of its neighboring nodes. For a given node $i$ and its associated neighborhood $N_i$, this aggregation is formalized as follows:

$$h_i = \sum_{j \in N_i} \mathbf{W}x_j. \tag{3}$$

This formulation enables GCNs to enhance the feature representation of each node by leveraging the attributes of its direct connections. However, given the variability in node degree, it is essential to normalize the aggregated features to ensure comparability across nodes. This normalization is achieved by factoring in the degree of the node, leading to the expression:

$$h_i = \frac{1}{\deg(i)} \sum_{j \in N_i} \mathbf{W}x_j. \tag{4}$$

Kipf et al. further refined the GCN architecture by addressing the potential imbalance in feature propagation, whereby nodes with a greater number of neighbors may disproportionately influence the learning process. To mitigate this effect, they proposed a weighted aggregation mechanism that accounts for the degrees of both the target node and its neighbors. The updated formulation is expressed as:

$$h_i = \sum_{j \in N_i} \frac{1}{\sqrt{\deg(i)\deg(j)}} \mathbf{W}x_j. \tag{5}$$

This enhancement promotes a more equitable distribution of influence among nodes, thereby ensuring that features from less-connected nodes are adequately considered.

The versatility of GCNs has led to their incorporation in various advanced frameworks, including Graph Attention Networks (GAT) (Velickovic et al., 2017) and Message Passing Neural Networks (MPNN). Their capacity to capture complex relational patterns and dependencies within graph structures renders GCNs particularly suited for applications spanning diverse domains, such as social network analysis, recommendation systems, and molecular property prediction in cheminformatics.

Additionally, GCNs can be further refined through modifications such as attention mechanisms that differentially weight the contributions of neighboring nodes based on learned significance or by integrating diverse edge types to enrich the contextual information. These adaptations contribute to the ongoing research aimed at enhancing GCN performance across a wide spectrum of graph-related tasks. In the context of our model, GCNs are instrumental in leveraging the structural information inherent in molecular graphs, facilitating improved predictive accuracy with respect to compound properties.

**Graph Isomorphism Network (GIN).** The Graph Isomorphism Network (GIN) is a neural network architecture introduced by Xu et al (Xu et al., 2018). in 2019 that aims to improve the expressive capabilities of graph neural networks (GNNs). GIN is particularly significant due to its equivalence to the Weisfeiler-Lehman (WL) graph isomorphism test, which serves as a standard for assessing the ability of models to distinguish between different graph structures.

The update mechanism for GIN aggregates node features and those of their neighbors using the following formulation:

$$h_v^{(k)} = \text{MLP}^{(k)} \left( (1 + \varepsilon) h_v^{(k-1)} + \sum_{u \in \mathcal{N}(v)} h_u^{(k-1)} \right) \tag{6}$$

In this equation, $h_v^{(k)}$ denotes the representation of node $v$ at the $k$-th layer, while $\mathcal{N}(v)$ represents the set of neighboring nodes. The term $\text{MLP}^{(k)}$ indicates a multi-layer perceptron applied to the aggregated features. The parameter $\epsilon$ is incorporated to preserve the unique identity of node features, thereby enhancing the model's ability to differentiate between nodes based on their characteristics.

GIN operates using a two-step framework: initially performing aggregation of neighboring features, followed by the application of a multi-layer perceptron. This approach facilitates the learning of complex representations that capture both local and relational information within graph structures.

Empirical evaluations of GIN demonstrate its superior performance in graph classification tasks compared to other GNN variants, underscoring its robustness across various datasets. The architecture coalesces well with applications where fine distinctions in graph structures are essential, such as in the prediction of molecular properties.

In this study, the integration of GIN into our model is anticipated to enhance the ability to capture intricate relationships within molecular graphs. This choice aims to improve the predictive performance across diverse physicochemical tasks, contributing to a more accurate assessment of chemical compounds.

**Graphormer.** Graphormer is an advanced architecture designed to enhance the capabilities of the Transformer model specifically for graph representation learning, as introduced by Ying et al. (Ying et al., 2021) This architecture effectively addresses the limitations encountered by traditional Transformer models, which often struggle to capture the inherent structural information present in graph data. To this end, Graphormer incorporates several innovative mechanisms, including centrality encoding, spatial encoding, and edge encoding, thereby improving the representation of graph data.

1. Centrality Encoding: Graphormer enhances the feature representation of nodes by integrating degree centrality into the input features. For a node $v$, the encoded feature is defined as:

$$h_v^{\text{centrality}} = h_v + \text{MLP}(\deg(v)), \tag{7}$$

where $h_v$ represents the original feature vector of node $v$, $\deg(v)$ denotes the degree of node $v$, and MLP denotes a multi-layer perceptron that transforms the centrality information into a vector space that aligns with the node features.

2. Spatial Encoding: The architecture utilizes spatial encoding to represent the shortest path distance (SPD) between nodes. The SPD between nodes $u$ and $v$ is computed and expressed as:

$$\text{spatial}(u, v) = \frac{1}{\text{SPD}(u, v) + 1},\tag{8}$$

where $\text{SPD}(u, v)$ denotes the shortest path distance between nodes $u$ and $v$.

3. Edge Encoding: To effectively utilize the significance of edge features, Graphormer incorporates edge encoding by calculating the interaction between edge features and node embeddings. This edge encoding is defined as:

$$e(u, v) = \frac{\text{dot}(h_u \cdot W_Q, h_v \cdot W_K)}{\sqrt{d}},\tag{9}$$

where $e(u, v)$ represents the embedded feature for the edge connecting nodes $u$ and $v$, $W_Q$ and $W_K$ are query and key martices respectively, d corresponds to the hidden dimension. This interaction is integrated into the attention mechanism by modifying the attention score as follows:

$$\text{Attention}(u, v) = \frac{\exp(e(u, v) + \text{spatial}(u, v))}{\sum_{w \in \mathcal{N}(u)} \exp(e(u, w) + \text{spatial}(u, w))} \cdot V,\tag{10}$$

where $\mathcal{N}(u)$ represents the set of neighbors of node $u$ and $V$ is value matrix.

Graphormer has exhibited state-of-the-art performance across a variety of graph-level tasks, including graph classification and molecular property prediction, demonstrating its versatility and robustness. By integrating Graphormer into our model, we leverage its advanced mechanisms to accurately capture intricate relationships and patterns within molecular graphs, significantly enhancing predictive performance across a broad spectrum of physicochemical tasks.

### A.3 TESTING DATASETS (QSAR)

**QM7.** The QM7 dataset is a curated subset of GDB-13, a comprehensive database containing nearly one billion stable and synthetically accessible organic molecules. Specifically, QM7 includes 7,165 molecules, each composed of up to 23 atoms, with a focus on seven heavy atoms: carbon (C), nitrogen (N), oxygen (O), and sulfur (S). This dataset not only provides a diverse array of molecular structures—such as double and triple bonds, cyclic compounds, carboxylic acids, cyanides, amides, alcohols, and epoxides—but also features the Coulomb matrix representation of these molecules. Additionally, the atomization energies for the QM7 molecules are computed using methods aligned with the FHI-AIMS implementation of the Perdew-Burke-Ernzerhof hybrid functional (PBE0).

**QM8.** The QM8 dataset consists of 21,786 small organic molecules and serves as a critical resource for evaluating machine learning models in predicting quantum mechanical properties. Each molecule is characterized by quantum chemical properties, including total energies and electronic spectra derived from time-dependent density functional theory (TDDFT). Although TDDFT offers favorable computational efficiency for predicting electronic spectra across chemical space, its accuracy can be limited.dataset is used to validate machine learning models in a prediction of deviations between TDDFT predictions and reference second-order approximate coupled-cluster (CC2) singles and doubles spectra. This approach has successfully applied to the low-lying singlet-singlet vertical electronic spectra of over 20,000 synthetically feasible small organic molecules.

**QM9.** The QM9 dataset is a prominent collection in computational chemistry, comprising 133,885 molecules with up to nine heavy atoms, including carbon (C), nitrogen (N), oxygen (O), and fluorine (F). This dataset is particularly valuable for evaluating machine learning models as it features a rich set of molecular structures representative of a wide chemical space.

Each molecule is identified by a unique 'gdb9' tag facilitating data extraction and a consecutive integer identifier (i). Rotational constants (A, B, and C, in GHz) describe the molecule's rotational inertia. The dipole moment ($\mu$, in Debye) indicates the molecule's polarity, while isotropic polarizability ($\alpha$, in $a^3$) reflects its response to electric fields. The energies of the highest occupied molecular orbital (HOMO) and lowest unoccupied molecular orbital (LUMO), both in Hartree (Ha), are included, along with the energy gap ($lumo - homo$, also in Ha). Electronic spatial extent ($R^2$, in Ha) characterizes the molecule's size. Vibrational properties are represented by the zero-point

vibrational energy ($zpve$, in Ha). Thermodynamic properties at 0 K and 298.15 K are also provided, including internal energy ($U_0$ and $U$, in Ha), enthalpy ($H$, in Ha), Gibbs free energy (G, in Ha), and heat capacity ($Cv$, in cal/mol K).

**FreeSolv.** The FreeSolv database is a comprehensive resource that offers a curated collection of experimental and calculated hydration-free energies for small neutral molecules in water. This database integrates both experimental values obtained from established literature and calculated values derived from advanced molecular dynamics simulations. It encompasses 643 small molecules, significantly expanding upon a previously existing dataset of 504 molecules. FreeSolv includes essential metadata, such as molecular structures, input files, and annotations, facilitating ease of access and reproducibility in research. The calculated values are derived from alchemical free energy calculations employing the Generalized Amber Force Field (GAFF) within a TIP3P water model, utilizing AM1-BCC charges. Calculations were conducted using the GROMACS simulation package, ensuring high accuracy and reliability. Furthermore, the database is regularly updated with new experimental references and data, enhancing its utility as a dynamic and evolving resource for the research community. Detailed construction processes and references are documented to provide transparency and context for users.

**ESOL.** The ESOL (Estimated SOLubility) dataset, introduced by Delaney ((Delaney, 2004)), provides a robust method for estimating the aqueous solubility of compounds directly from their molecular structure. The model, derived from a comprehensive training set of 2,874 measured solubilities, employs linear regression analysis based on nine molecular properties, with calculated logP octanol identified as the most significant parameter. Other key descriptors include molecular weight, the proportion of heavy atoms in aromatic systems, and the number of rotatable bonds. ESOL demonstrates competitive performance relative to the well-established General Solubility Equation, particularly for medicinal and agrochemical compounds. In our study, we build upon the ESOL dataset by utilizing a superstructure aimed at predicting water solubility across an extended set of 1,128 samples. This enhancement not only broadens the applicability of the original model but also supports more precise solubility estimations in diverse chemical spaces. The combination of ESOL's foundational framework with our superstructure facilitates further exploration of solubility-related properties, making it a valuable tool for researchers in drug discovery and environmental sciences.

**LIPO (Lipophilicity).** The lipophilicity dataset is a vital resource for examining the pharmacokinetic properties of drug molecules, specifically in relation to membrane permeability and solubility. Curated from the ChEMBL database, this dataset encompasses experimental results for the octanol/water distribution coefficient (logD) at pH 7.4 across a diverse collection of 4,200 compounds. Lipophilicity, described by the n-octanol/water partition coefficient or the n-octanol/buffer solution distribution coefficient, is of considerable significance in pharmacology, toxicology, and medicinal chemistry. In this study, a quantitative structure–property relationship (QSPR) analysis was conducted to predict logD values at pH 7.4 for the dataset. Comparative analysis with previously established logD values demonstrated that the developed predictive model offers reliable and robust performance. This enhances its utility as a valuable tool for researchers aiming to evaluate and optimize the lipophilicity of potential drug candidates, thereby informing pharmacological strategies in drug development.

**BBBP.** The Blood-Brain Barrier Permeability (BBBP) dataset serves as a resource for studying the ability of chemical compounds to penetrate the blood-brain barrier (BBB), which is an important consideration in drug development for central nervous system disorders. The BBB selectively regulates the transfer of substances from the bloodstream into the brain, thereby necessitating an accurate assessment of BBB penetration for potential therapeutic agents. In this study, the original BBBP dataset was modified to create both free-form and in-blood-form datasets. Molecular descriptors were generated for each dataset and employed in machine learning (ML) models to predict BBB penetration. The dataset was partitioned into training, validation, and test sets using the scaffold split algorithm from MoleculeNet, which intentionally creates an unbalanced partition to enhance the evaluation of predictive performance for compounds that are structurally dissimilar to those used in the training data. Notably, the random forest model achieved the highest prediction score using 212 descriptors from the free-form dataset, surpassing previous benchmarks derived from the same splitting method without any external database augmentations. Additionally, a deep

neural network produced comparable results with just 11 descriptors, emphasizing the significance of recognizing glucose-like characteristics in the prediction of BBB permeability.

**Tox21.** The Tox21 dataset is a significant resource in toxicology research, comprising 12,060 training samples and 647 test samples representing various chemical compounds. Each sample is associated with 12 binary labels reflecting the outcomes (active/inactive) of different toxicological experiments, although the label matrix contains numerous missing values. Due to the extensive size of the dataset, our study focuses exclusively on predicting the NR-AR property. Since its inception in 2009, the Tox21 project has screened approximately 8,500 chemicals across more than 70 high-throughput assays, yielding over 100 million data points, all publicly accessible through partner organizations such as the United States Environmental Protection Agency (EPA), National Center for Advancing Translational Sciences (NCATS), and National Toxicology Program (NTP). This collaborative effort has produced the largest compound library specifically aimed at enhancing understanding of the chemical basis of toxicity across research and regulatory domains. Each federal partner contributed specialized resources, culminating in a diverse set of compound libraries that collectively expand coverage of chemical structures, use categories, and properties. The integrated approach of Tox21 enables comprehensive analysis of structure–activity relationships through ToxPrint chemotypes, allowing the identification of activity patterns that might otherwise remain undetected. This dataset underscores the central premise of the Tox21 program: that collaborative merging of distinct compound libraries yields greater insights than could be achieved in isolation.

**ClinTox.** The ClinTox dataset serves as an a resource for understanding the factors influencing drug approval and toxicity outcomes in clinical trials. This dataset compares drugs approved by the FDA with those that have failed clinical trials due to toxicity reasons, encompassing two classification tasks for 1,491 drug compounds with known chemical structures. Specifically, it aims to classify (1) clinical trial toxicity (or absence of toxicity) and (2) FDA approval status. The compilation of FDA-approved drugs is derived from the SWEETLEAD database, while information regarding compounds that failed clinical trials is sourced from the Aggregate Analysis of Clinical Trials (AACT) database.

**BACE.** The BACE dataset is a resource for the study of inhibitors targeting human $\beta$-secretase 1 (BACE-1), a key enzyme involved in the pathogenesis of Alzheimer's disease. This dataset provides both quantitative binding results (IC50 values) and qualitative outcomes (binary labels) for a collection of 1,522 compounds, encompassing experimental values reported in the scientific literature over the past decade. Notably, some of these compounds have detailed crystal structures available, which enhances the dataset's utility for structure-activity relationship (SAR) studies. The BACE dataset has been integrated into MoleculeNet, where it is structured as a classification task, effectively merging the compounds with their corresponding 2D structures and binary labels. The use of scaffold splitting in this context is particularly beneficial, facilitating the assessment of predictive performance on a single protein target by preventing bias associated with structural similarities among compounds. This integration of experimental binding data and diverse structural information underscores the dataset's potential to aid in the design and optimization of BACE-1 inhibitors, ultimately contributing to advancements in therapeutic strategies for Alzheimer's disease.

**MUV.** The Maximum Unbiased Validation (MUV) dataset serves as a benchmark for evaluating virtual screening techniques in drug discovery. Selected from the PubChem BioAssay database, the MUV dataset comprises 17 challenging tasks associated with approximately 90,000 chemical compounds, strategically designed to facilitate robust validation of virtual screening methodologies. A key feature of this dataset is its foundation in refined nearest neighbor analysis, a technique derived from spatial statistics that offers a mathematical framework for the nonparametric analysis of mapped point patterns. This methodology enables the systematic design of benchmark datasets by purging compounds that exhibit activity against pharmaceutically relevant targets while eliminating unselective hits. Through topological optimization and experimental design strategies, the refined nearest neighbor analysis constructs data sets of active compounds and decoys, ensuring they are unbiased concerning analogue bias and artificial enrichment. Consequently, the MUV dataset provides an essential resource for Maximum Unbiased Validation, empowering researchers to assess and improve the predictive performance of virtual screening methods in a more rigorous manner.

**HIV.** The HIV dataset, introduced by the Drug Therapeutics Program (DTP) AIDS Antiviral Screen, encompasses an extensive screening of over 40,000 compounds to assess their inhibitory effects on HIV replication. The screening results are categorized into three classifications: confirmed inactive (CI), confirmed active (CA), and confirmed moderately active (CM). For the purposes of analysis, CA and CM labels are combined to formulate a binary classification task distinguishing between inactive (CI) and active (CA/CM) compounds. This dataset is particularly valuable for researchers aiming to discover new categories of HIV inhibitors, and the use of scaffold splitting is recommended to enhance the identification of novel compounds while mitigating bias related to structural similarities. Additionally, the HIV positive selection mutation database provides a comprehensive resource for understanding the selection pressures exerted on HIV protease and reverse transcriptase, which are critical targets for antiretroviral therapy. This large-scale database contains sequences from approximately 50,000 clinical AIDS samples, leveraging contributions from Specialty Laboratories, Inc., allowing for high-resolution selection pressure mapping. It offers insights into selection pressures at individual sites and their interdependencies, along with datasets from other public repositories, such as the Stanford HIV database. This confluence of data facilitates cross-validation with independent datasets and enables a nuanced evaluation of drug treatment effects, significantly advancing the understanding of HIV resistance mechanisms.

## A.4    SOME TRAINING DETAILS

**Weighted Cross-Entropy Loss.**    Weighted cross-entropy loss assigns different weights to different classes based on their frequency in the dataset. Such approach is useful when you have unbalanced data and you want the model to pay more attention to less represented classes. Class weights do compensate for the imbalance by increasing the contribution of rare classes to the total loss, according to the formulae:

$$L = -\frac{1}{N} \sum_{i=1}^{N} \sum_{c=1}^{C} w_c \cdot y_{i,c} \cdot \log(p_{i,c} + \epsilon), \tag{11}$$

where
- $N$ - the number of examples in the batches,
- $C$ - number of classes,
- $w_c$ - weight for class $c$,
- $y_{i,c}$ - true label for example $i$ and class $c$,
- $p_{i,c}$ - probability predicted by the model for example $i$ and class $c$ (after applying softmax),
- $\epsilon$ - a small value to prevent division by zero.

This formulae calculates the average of the weighted cross-entropy over all examples in the batches. We used this variation of Cross-Entropy Loss for the HIV, the Tox21, the ClinTox and the MUV datasets to improve the quality of our models.

