# OpenReview forum: "Broadening Discovery through Structural Models: Multimodal Combination of Local and Structural Properties for Predicting Chemical Features."
_ICLR.cc/2025/Conference — ICLR 2025 Conference Withdrawn Submission_

### Official Review · Reviewer_tjx6 · 2024-10-15

**Soundness:** 2
**Presentation:** 3
**Contribution:** 2
**Rating:** 3
**Confidence:** 5

**Summary:**

This study presents two major contributions: 1. the use of extended connectivity fingerprint hashes as tokens for training a molecular language model (as opposed to the current paradigm using SMILES characters as tokens); 2. the use of a bimodal architecture for pre-training combining the aforementioned language model with a graph neural network architecture to provide a combined molecular representation.

The study considers the use of three different GNN architectures, namely, Graph Convolutional Network (GCN), Graph Isomorphism Network (GIN), and Graphformer.

Evaluation is performed on the MoleculeNet benchmark datasets describing biophysical properties of small molecules and prediction of quantum properties. The results show 1. that the language model outperforms another language model (i.e., ChemBERTa), 2. that the bimodal model combining LM with GNN outperforms the baselines herein considered, 3. that the GIN architecture is better suited for smaller datasets, whereas the Graphformer is the best option for bigger datasets.

**Strengths:**

1. The idea of using ECFP hashes as tokens for language modelling is an elegant solution to the problems with SMILES-based language models and introduces a much needed chemical inductive bias to the modelling paradigm. The comparison between ECFP-BERT and ChemBERTa seem to support this conclusion with the former outperforming the latter in 8 out of 11 datasets.
2. The idea of only masking atom and edge types during pretraining of the graph models makes a lot of intuitive sense and is an interesting incremental contribution to previous efforts (lines 427-431 and Figure 6).
3. The insight as to what GNN architecture might be a better choice depending on the dataset size is also an interesting finding (lines 481-485).
4. The paper is written clearly and concisely, main claims and contributions are succinctly expressed. The analysis and discussion are well written.

**Weaknesses:**

Main weaknesses
-----
1. There is an important methodological inconsistency regarding the nature of the pre-training dataset. The methods section (line 265) that "[the RoBERTa architecture] has been trained on ECFPs derived from the PubChem and ZINC datasets [...]"; however, the experiments section (line 457) contradicts this statement: "The three first models [RoBERTa, RoBERTa+GIN, RoBERTA+GCN] utilized a dataset comprising 10 million entries sourced from the PubChem database [...]". The authors should clarify what the pre-training dataset is and resolve this inconsistency.
2. The baselines in Tables 1 and 2 are insufficient. One of the main claims of the paper is that the ECFP-BERT approach (RoBERTa trained on ECFP hashes) enhances performance, however, the only alternative LM considered is ChemBERTa which is significantly outdated with ChemBERTa-2 already available [1], as well as, other alternatives like MolFormer-XL [2] which has reported better performance in some of the datasets considered (BBBP, Tox21, ClinTox, HIV, FreeSol, Lipo, ESOL) than the performance here reported for ECFP-BERT (Table 1 and 2 in [2]). Furthermore, the performance of ECFP fingerprints alone with either a MLP or other traditional ML downstream model (SVM, Random Forest, LightGBM, XGBoost, etc.) should have been included to demonstrate the ECFP-BERT approach is enhancing performance as opposed to directly using the fingerprints themselves.
3. This is a minor issue compared to the two previous points, but the values in Tables 1 and 2 should by accompanied by a measure of the error across different runs to get an idea of the stability of the training and the statistical significance of the differences between models.
4. An ablation study should be conducted where the performance of the pre-trained GCN, GIN, and Graphformer architectures alone without BERT, can be evaluated. It would also be illustrative to include an example where a transformer is trained in SMILES substituting ECFP-BERT, to properly demonstrate the superior performance of the approach presented in the paper.

Minor points
----
_(These points do not have a direct bearing on my decision regarding the score of the paper, but I feel they could improve the paper)_
1. Typos:
   - a. Line 105: "we **it** implement". The "it" should be removed.
   - b. Line 140-141: "This sensible of surrounding environment representations". The expression is unclear and I cannot infer the meaning it is trying to convey.
   - c. Line 200: "The language model is designed to accept Extended Connectivity Fingerprint (ECFP) **connectives**. I think that a better choice would have been **substructures** from the chemical point of view or, alternatively, for a computer science audience, **hashes**.
   - d. Line 209: "(approximately from **-2^32 to 2^32**)". I have assumed that this refers to "0 - 2^32".
    - e. Line 214: "the largest dataset we have - PubChem, containing 10M molecules". This ties with the 1st Major weakness as if ZINC was also used, then ZINC is larger (~37B molecules [3]).
2. Regarding the comparison with Molformer-XL, that model was trained on 10% of the ZINC database, so perhaps ECFP-BERT would perform better with a larger pre-training dataset.
3. In Table 1, the values for BERT-GIN and BERT-GCN are both 0.79 for Tox21, but only the BERT-GIN is bold.
4. As one of the conclusions of the paper is that BERT-GIN is a better option than BERT-Graphformer for smaller datasets and for bigger datasets, the opposite is true, it would be helpful to have a Table (for example in the Appendix) describing the size of each of the datasets, to better support this argument.


References
----
1. Ahmad W, Simon E, Chithrananda S, Grand G, Ramsundar B. Chemberta-2: Towards chemical foundation models. arXiv preprint arXiv:2209.01712. 2022 Sep 5.
2. Ross J, Belgodere B, Chenthamarakshan V, Padhi I, Mroueh Y, Das P. Large-scale chemical language representations capture molecular structure and properties. Nature Machine Intelligence. 2022 Dec;4(12):1256-64.
3. Tingle BI, Tang KG, Castanon M, Gutierrez JJ, Khurelbaatar M, Dandarchuluun C, Moroz YS, Irwin JJ. ZINC-22─ A free multi-billion-scale database of tangible compounds for ligand discovery. Journal of chemical information and modeling. 2023 Feb 15;63(4):1166-76.

**Questions:**

1. The ECFP fingerprints depend on the radius used to calculate it, as pointed out in related works (line 140); I haven't found the radius used either in the main text or the appendix. If I have missed it, could you point me towards where it is stated? Otherwise, what is the value?

**Details Of Ethics Concerns:**

The model describes an application of AI for Science, where the training data is publicly available and do not contain any personal or sensitive information. There is no risk for societal harm from the consequences of this research.

---

### Official Review · Reviewer_RWv5 · 2024-10-16

**Soundness:** 1
**Presentation:** 2
**Contribution:** 1
**Rating:** 1
**Confidence:** 4

**Summary:**

This study investigates the potential of language models based on fingerprints within a bimodal architecture that combines both graph-based contrastive learning and a language model using ECFP as input. The evaluation includes several model architectures: RoBERTa with ECFP input, contrastive learning to align RoBERTa with ECFP input, and GCN/GIN/Graphformer models. They test the capacity of their pre-trained model on several tasks from the MoleculeNet benchmark.

**Strengths:**

The paper addresses an important problem in molecular representation, which is a timely and relevant area of research.

**Weaknesses:**

RoBERTa model:
1. Inappropriateness of ECFP for Language Models:
- Lack of Sequential Information: A fundamental issue with this approach is the mismatch between the unordered nature of ECFP and the ordered nature of language models like RoBERTa. ECFP fingerprints are essentially sets of hashed molecular substructures that have no inherent order or sequence. Thus, ECFP fingerprints capture local molecular environments around atoms, but they do not preserve topological information—the connectivity between atoms and the overall molecular structure. By using a language model, which assumes token sequences have some structural or positional significance, this model risks ignoring the important chemical graph structure of the molecule. In molecules, the way atoms are connected is critical to their chemical properties, and ECFP alone does not convey this information.
- Use of BPE Tokenization on ECFP Hashes: Fragmenting a hash into smaller tokens using BPE will result in the loss of critical information. Hashes are designed to be compact representations, and breaking them down leads to uninformative tokens.
- Masked Token Prediction as Self-Supervised Task: The paper uses masked token prediction as a self-supervised task, which is very effective in NLP where there is sequential context. However, predicting masked ECFP tokens is less intuitive because there is no inherent relationship between neighboring ECFP hashes in the fingerprint.

GNN models:
The authors employ contrastive learning for the GNN model to learn structural information about molecules by creating augmented graphs through atom masking. They aim to use contrastive loss to ensure that representations of augmented graphs from the same molecule are close in embedding space, while those from different molecules are farther apart. However, this approach may be ineffective because different molecules have varying graph structures, resulting in naturally greater distances between positive-negative pairs than between positive-positive pairs. Consequently, the parameter optimization process may not be effective, as this could be considered an easy problem. Additionally, the paper lacks a clear description of the graph augmentation methods, raising questions about the validity of their approach.

Downstream task benchmarking:
The authors do not provide information on how they split the data in the MoleculeNet dataset, raising concerns about the comparability and reliability of the reported results.

**Questions:**

Data Splitting: Can you clarify how you split the MoleculeNet dataset for your experiments?

Graph Augmentation: Could you provide more details on the graph augmentation methods employed in your study?

---

### Official Review · Reviewer_9ueJ · 2024-11-03

**Soundness:** 2
**Presentation:** 3
**Contribution:** 3
**Rating:** 5
**Confidence:** 4

**Summary:**

This article introduces a multimodal architecture combining graph neural networks with an ECFP-based RoBERTa language model, enhancing chemical property predictions across different tasks. Keys contribution include:
1. Introducing different views of molecules for better representation learning
2. Performing data augmentation by masking different atom and bond information, and utilize constrastive learning to minimize the discrepancy between different graphs.

**Strengths:**

1. The combination of graph neural networks with a language model based on ECFP is novel.
2. The method outperforms other baseline methods include MolCLR.

**Weaknesses:**

The main problem of the paper is about the baseline selection.
The authors cited the "Dual-view Molecular Pre-training" (DVMP) paper but didn't use it as a baseline. The main difference between this paper and the DVMP paper is, that paper use the SMILES and molecule graph as input, trying to maintain dual-view consistency between these two formats. While this paper change the SMILES to ECFP, takes an unconventional approach by forcefully tokenizing ECFP representations and then pre-training them in a language model. I'm not sure whether this is suitable or not because I don't see other papers have done this before, but if the authors want to prove the necessity of using ECFP over SMILES for a language model, they should use the DVMP method as a baseline.

**Questions:**

See above

---

### Note · Authors · 2024-11-26

I have read and agree with the venue's withdrawal policy on behalf of myself and my co-authors.